# Outcome Prognostic Factors in MRI during Spica Cast Therapy Treating Developmental Hip Dysplasia with Midterm Follow-Up

**DOI:** 10.3390/children9071010

**Published:** 2022-07-07

**Authors:** Katharina Susanne Gather, Ivan Mavrev, Simone Gantz, Thomas Dreher, Sébastien Hagmann, Nicholas Andreas Beckmann

**Affiliations:** 1Clinic for Orthopedics and Trauma Surgery, Center for Orthopedics, Trauma Surgery and Spinal Cord Injury, Heidelberg University Hospital, Schlierbacher Landstrasse 200a, 69118 Heidelberg, Germany; ivan.mavrev@gmx.de (I.M.); simone.gantz@med.uni-heidelberg.de (S.G.); sebastien.hagmann@med.uni-heidelberg.de (S.H.); nicholas.beckmann@med.uni-heidelberg.de (N.A.B.); 2Pediatric Orthopedics and Traumatology, Children’s University Hospital Zurich, Steinwiesstrasse 75, 8032 Zurich, Switzerland; thomas.dreher@kispi.uzh.ch

**Keywords:** spica cast, closed reduction, developmental hip dysplasia, hip luxation, MRI

## Abstract

Closed reduction followed by spica casting is a conservative treatment for developmental dysplasia of the hip (DDH). Magnetic resonance imaging (MRI) can verify proper closed reduction of the dysplastic hip. Our aim was to find prognostic factors in the first MRI to predict the possible outcome of the initial treatment success by means of ultrasound monitoring according to Graf and the further development of the hip dysplasia or risk of recurrence in the radiological follow-up examinations. A total of 48 patients (96 hips) with DDH on at least one side, and who were treated with closed reduction and spica cast were included in this retrospective cohort study. Treatment began at a mean age of 9.9 weeks. The children were followed for 47.4 months on average. We performed closed reduction and spica casting under general balanced anaesthesia. This was directly followed by MRI to control the position/reduction of the femoral head without anaesthesia. The following parameters were measured in the MRI: hip abduction angle, coronal, anterior and posterior bony axial acetabular angles and pelvic width. A Graf alpha angle of at least 60° was considered successful. In the radiological follow-up controls, we evaluated for residual dysplasia or recurrence. In our cohort, we only found the abduction angle to be an influencing factor for improvement of the DDH. No other prognostic factors in MRI measurements, such as gender, age at time of the first spica cast, or treatment involving overhead extension were found to be predictive of mid-term outcomes. This may, however, be due to the relatively small number of treatment failures.

## 1. Introduction

The estimated incidence of developmental dysplasia of the hip (DDH) in infants varies geographically from 0.1–30 per 1000 newborns, depending on the population [1,2,3]. Unrecognized and untreated DDH can lead to premature osteoarthritis and is responsible for up to one-third of hip replacements in adults younger than 60 years [1]. The exact etiology of DDH remains unknown and is likely multifactorial, with its pathophysiology reflecting a combination of primary abnormal acetabular development and secondary abnormal interaction between the femoral head and the acetabulum during perinatal life [4]. Specifically, previous studies have shown that breech intrauterine position, family history of DDH and female gender are the most important risk factors for DDH [1,5]. Other risk factors such as first-born, oligohydramnios, overly restrictive swaddling practices and foot abnormalities have been linked to DDH, but the evidence is weaker [2,5].

Detected at an early age by ultrasound, developmental hip dysplasia (DDH) can be very effectively treated by conservative means, even in severe cases. Success rates of 90.4–99.8% have been shown for the Tübinger splint, the Pavlik harness, or casts, and as such are considered first-line therapy in neonates [6,7,8,9]. These splints or casts exploit the huge potential for growth and remodeling of the newborn hip [10]. The goal of treatment in these cases is to achieve and maintain concentric reduction of the hip joint to promote femoral head coverage, and congruent acetabular and femoral head development. Treatment is guided by the age at presentation and the severity of the disease [11]. Later, once the ossification center has formed in the femoral head, the sonographic possibilities become limited and AP X-ray of the pelvis provides superior diagnostic information [12]. Here, the acetabular index is the relevant parameter up to the age of 4. Beginning at the age of 4, the centre-edge angle (CE angle), according to Wiberg, becomes increasingly relevant [13].

In German-speaking countries, conservative therapy is usually guided by ultrasound of the hip. In these regions, the Graf technique has prevailed (see also Section 2.3 Measurements).

The algorithm varies in the case of spica casting. Some prefer leaving window in the cast to perform a transinguinal ultrasound [14,15,16]. This additional window reduces the stability of the cast, which is why others prefer magnetic resonance imaging (MRI) to evaluate the position of the femoral head after closed reduction and cast application [17,18,19,20,21]. Further available imaging methods are X-ray or computer tomography (CT) scans [22,23,24]. Reported rates of recurrent dislocation identified on cross-sectional imaging after closed reduction range from 6% to 15% [12]. Although the sensitivity and specificity of CT and MRI for detecting dislocation have been found to be equivalent [25], MRI has the advantage of not exposing the sensitive developing tissues to ionizing radiation and providing superior soft-tissue resolution and enhancement profiles. Specifically, in cases of abnormal post-reduction hips, MRI can identify obstacles to reduce and detect unexpected complications [12]. Due to the unnecessary radiation exposure and no proper display of the cartilaginous hip, we prefer MRI after closed reduction and spica cast application. Even though the correct centre of the femoral head is the main concern, several indices can be measured in MRI. Jaremko et al. evaluated several of these for dysplasia on infant hip MRI (see Section 2.3 Measurements) [26]. Indices of hip dysplasia and adequacy of reduction differ between modalities, including ultrasound, radiography, CT, MRI and arthrography with limited cross-correlation. Jaremko comprehensively adapted all available DDH indices from CT and other modalities to MRI, and tested which could be feasibly measured on MRI, assessed interobserver variability, and correlated indices to each other [26].

Using these indices, our aim was to find parameters on the first MRI after closed reduction and spica cast application correlating with the outcome. In this regard, we looked for prognostic factors predicting the expected success of treatment. To our knowledge, there are no comparable studies to this topic so far.

By means of the parameters measurable in MRI, we wanted to find out whether they can predict conservative therapy success or failure in terms of residual dysplasia or recurrence rate.

## 2. Patients and Methods

### 2.1. Patients Population

In this retrospective cohort study, we included all 48 patients (96 hips) with DDH on at least on one side, who were consecutively treated with closed reduction and spica cast between 2005 and 2016 at our institution. The diagnosis was confirmed by an ultrasound in our department. Moreover, 84/96 (87.5%) were female and 12 (12.5%) were male; 54.2% (26/48) were bilateral and 45.8% (22/48) had DDH on one side. Since a spica cast always retains both hips, the healthy opposite sides were also retained in the patients affected on only one side—no reduction was necessary, nor occurred on the healthy side. This ultimately resulted in 69 hips with DDH and 27 healthy hips being treated with a spica cast.

Initial Graf types were type 1a or 1b 27/96 (27.2%), type 2a 9/96 (9.1%), type 2b 3/96 (3.0%), type 2c 12/96 (12.1%), type D 12/96 (12.1%), type 3a 21/96 (21.2%), type 3b 3/96 (3.0%) and type 4 7/96 (7.1%) [27,28,29]. One patient had no initial ultrasound in our data. Six patients (12.5%) had received overhead extension prior to spica casting. Furthermore, 3/6 had at least on one site Graf type 4, 1 type 3b and 1 patient had a type 2c hip at initial presentation to our outpatient department after having been treated with overhead extension and spica cast in different hospitals before (the hip was initially Graf type 4). The remaining patient had already started ossification of the unstable decentred femoral head upon initial presentation to our outpatient department at the age of 5 months. We initiated an overhead extension because of the age of the child and the instability of the hip.

Patients with neurogenic or syndromic hip dysplasia were excluded. The exact age at diagnosis could not be determined adequately because the records are incomplete. We were not able to retrospectively verify the time of initial diagnosis for each patient, which was generally made prior to presentation to our department. The treatment began at our department at a mean child age of 9.9 weeks (s = 6.0) (range 4 to 33 weeks). Treatment of male patients started at a mean of 7.3 weeks (s = 2.2) and female patients at 10.3 weeks (s = 6.3) of age. This difference was highly significant (*p* = 0.003). A follow-up was 47.4 months on average (s = 34.5). 9/48 (18.8%) children had a known risk factors for DDH with a breech presentation and/or positive family history.

### 2.2. Treatment Technique

All patients received a Graf hip ultrasound at our institution to confirm the diagnosis. In cases of an unstable hip (type IIc unstable or more according to Graf’s classification) on at least one side, we performed closed reduction and spica casting (see Figure 1 and Figure 2). The patients were deemed too old to try a Tübingen splint when they presented to our outpatient clinic. Cooper et al. and Seidl et al. described the Tübingen splint as a successful option for very young infants. The standard therapy for unstable hips at this age is the spica cast [30,31]. The duration of the cast therapy was determined according to the patient’s age and the severity of the disease. The standard therapy was two serial casts of 3 weeks duration. Afterwards, an ultrasound reevaluation was performed. If the DDH was still severe, a further spica cast was applied again for an additional 3 weeks if necessary. If DDH was deemed mild, or the alpha angle was less than the target of 64° or more, a Tübingen splint was applied; if an alpha angle of 64° or more was measured, the treatment was stopped. If a severe DDH was initially present in a very young infant, a 3-week cast therapy was sometimes carried out and then an ultrasound reevaluation performed to decide if the standard two rounds of 3-week treatment spica cast should be continued, or if another option, such as a Tübingen splint was sufficient.

To reach a good result, the reduction was performed under general balanced anaesthesia with laryngeal masks. In all cases, MRI was performed directly after closed reduction to control the position/reduction of the femoral head without additional anaesthesia by taking advantage of residual anaesthesia. In unsteady infants, positioning aids were used, such as a light sandbag to support the spica cast, to additionally minimize movement artefacts in MRI. For MRI sequences see Table 1. The spica cast was renewed after 3 weeks once or twice depending on the initial severity of DDH.

After the scheduled spica cast therapy, an ultrasound was carried out to reevaluate the hip. If the alpha angle was still below 64°, the therapy was continued with a Tübingen splint until the desired maturation of the hip occurred. The Tübingen hip flexion splint is a removable orthosis that fixes the squat position of the hip and was invented by Prof. Bernau for the treatment of stable hip joints [32]. In this study, treatment control was performed using ultrasound with the technique described by Graf. An alpha angle of at least 60° (Graf Typ I) was required to end therapy (treatment goal was defined as an alpha angle over 64°, but in cases where a type 1 hip had been achieved some parents opted to end further treatment). Afterwards, the children were followed-up according to our in-house standards by means of X-ray until the end of growth. The patients presented 2–3 months after the start of walking, around the age of 3, shortly before starting school, before the pubertal growth spurt and at the end of growth (see Figure 1). This way, recurrences could be detected accordingly and the development of the hip could be observed. Mild dysplasia was further observed and partly treated with a nighttime abduction wedge. If the CE angle was less than 15°, surgical reconstruction was indicated.

### 2.3. Measurements

Ultrasound was performed with the Graf technique [33,34]. The child is placed in a lateral position in a positioning tray and examined with a linear ultrasound transducer. This is applied to the greater trochanter in the longitudinal axis of the child and produces a frontal section through the acetabulum. After anatomical identification and usability testing, the α- and β-angles are determined (see Figure 3). The tangent to the os ilium is the baseline for both angle determinations. Two guide lines are added to this baseline to measure the bony acetabular roof angle α (bone angle) and the cartilaginous acetabular roof angle β (cartilage angle). For the latter, a connecting line (display line) is drawn between the bony acetabular notch (turnover point) and the centre of the acetabular labrum. The bony notch is located at the point where the acetabulum changes its profile from concave to convex. For the α-angle, a tangent is drawn to the bony acetabulum starting from the lower edge of the os ilium.

Alpha and beta angle were measured initially and after ending spica cast therapy, and until ending treatment or ossification of the femoral head. If the femoral head already ossified during follow-up we performed X-rays instead of ultrasound to follow-up. This usually happens around 8 months of age. 6 children were lost to follow-up at the first planned X-ray after they had begun walking. All other children showed femoral head ossification by the follow-up time. An angle of 64° or more was rated as excellent, 60–63.9 as good, and below this as poor. The aim was to achieve a type I hip according to Graf, which accordingly has an alpha angle of at least 60° and thus to achieve a normal hip.

For MRI indices, we used those described by Jaremko et al. (see Figure 4) [26]. First the hip abduction angle (Abd.) is measured in the axial T1 phase on the image with the largest diameter of the femoral heads. This is defined as the angle between a perpendicular line to Hilgenreiner’s line and the line along the mid femoral shaft. The coronal acetabular angle (CorAcet) is measured on the coronal image on which the acetabular roof is steepest. This image must also contain a substantial portion of the femoral head. CorAcet is defined as the angle between Hilgenreiner’s line and a line to the superior bony edge of the acetabulum. For the Pelvic width (PelvWid), we used the image showing the widest distance between medial ischial walls at the pelvic inlet. The width is a measure of the inner distance between the inner edge of left and right cortex. Similarly Jaremko et al., we used the axial image on which the acetabulum appears deepest to measure the anterior and posterior bony axial acetabular angles (AxAcet and AxPAcet). This image also contained a substantial part of the femoral head. The AxAcet is the angle between Hilgenreiner’s line and a line joining the anterior edge of the bony acetabulum to the lateral edge of the triradiate cartilage. The AxPAcet is measured similarly at the posterior joint line [26].

After ending spica casting, further follow-up was performed using X-ray. Children with DDH should be followed-up radiologically until growth completion, due to late recurrence. If no signs of DDH are present during follow-up, further follow-up at walking age, after 8–10 years and after the end of growth is recommended [35]. In our cases, severe DDH was present, so spica cast therapy was necessary and therefore the control intervals were reduced to walking age, again after 3 and 5 years, and after 10 years. Due to the study period and follow-up time frame, no patient has reached skeletal maturity at this point.

The centre edge angle as described by Wiberg (CE-angle) and the Acetabular Index (AI) were measured and we evaluated for further pathology, such as signs of beginning avascular femoral head necrosis. Although the CE-angle is not validated for children under 4 years age, it was used for comparison with other authors. Of particular interest was the further development of the hips and the femoral head coverage after successful post-maturation: Do the children develop a recurrence of hip dysplasia or do they remain mature without signs of bony DDH? This was evaluated and staged using the AC and CE angle relative to the patient age according to Tönnis [33].

All MRI and radiographic measurements were performed with TraumaCad Version 2.5 (Brainlab Ltd., Petah Tikva, Israel).

### 2.4. Statistical Analysis

All of the statistical analyses were conducted using SPSS (version 28; SPSS; Chicago, IL, USA). Descriptive statistics, including arithmetic mean value and standard deviation, were calculated. Data is given as mean ± standard deviation (SD) and ranges, if not indicated otherwise.

*T*-Tests and Pearson’s chi-squared-tests were used for group statistics. Prognostic factors were calculated with linear regression tests. Differences were considered significant if the *p*-value was <0.05.

## 3. Results

The mean duration of treatment was 11.3 weeks (S = 4.9). The treatment involved a mean of 6.0 weeks (3–9 weeks) cast and 5.3 weeks with Tübinger splint. There was no significant difference between male and female patients. After removing the cast, 70.8% (68/96) needed additional therapy with Tübinger splint depending on sonographic measurements. The others already reached an alpha angle of >64° after cast therapy and did not need any additional splint. These patients were primarily unilaterally affected infants. For initial Graf type see Table 2.

There was a positive correlation between having had an overhead extension and the need of a Tübinger splint after spica cast (*p* = 0.014).

One patient (both hips) needed surgical treatment during the follow-up period. A total of 3 patients (3 hips) still have a residual dysplasia and may require additional surgery at a later date, such as acetabuloplasty and derotation-varisation-osteotomy (DVO). The following measured parameters were not associated with surgery (either performed or recommended): gender, age at time of starting therapy, Graf Type, alpha-angle at the end of therapy or MRI measurements.

### 3.1. Sonographic Evaluation

The sonographic alpha- and beta-angle measured at the beginning and end of therapy separated by sex are summarized in Table 3. The difference of both angles (alpha and beta) was highly significant with *p* < 0.001. No significant correlation was noted between the first alpha-angle and the first radiologic measurements (CE-angle, AI, ACM, HAS). The initial Graf Type had no influence on the duration of the therapy in total, nor on further surgical treatment. However, the duration of spica cast therapy positively influences the alpha angle at the end of therapy (*p* = 0.003, beta = 0.869). No other measured parameter was shown to have a significant positive correlation. An alpha angle of 64° or more was rated as excellent, 60–63.9 as good, and below as poor.

No significant difference in alpha-angle or Graf Type was noted between the male and female patients. There was a difference in age at time of starting the therapy between both sexes (*p* = 0.003). Girls were treated on average 3 weeks later than boys. No significant differences were found in duration of the therapy (*p* = 0.646) or the alpha angle at the end of therapy.

A total of 3 hips in 2 patients (2/48, 4.1%) did not reach an alpha angle of 60° at the end of therapy. One was getting too old to continue immobilisation with regard to the motor development. The parents of the other patient declined further treatment with a spica cast or Tübinger splint to finish treatment.

### 3.2. Magnetic Resonance Imaging Measurements

Table 4 gives a summary of the measurements shown in the MRI. 31/48 (64.6%) patients had a second MRI, which was carried out after receiving the second spica cast. All of the MRI scans showed a concentric reduction of the femoral head within the acetabulum.

There was no significant difference of the measured angles depending on sex. There was a positive correlation between pelvic width and age at time of first spica cast (*p* < 0.001).

All of the measured parameters had no significant influence on the alpha angle at the end of therapy (outcome). Two parameters influenced duration of therapy, first AxPAcet (*p* = 0.028, beta = −0.218) and the Abduction (*p* = 0.003, beta = −0.304) in the first MRI.

### 3.3. Radiographic Evaluation

Two patients had no initial ultrasound examination because of age at time of presentation and an already ossified femoral head (25 and 33 weeks). During further follow-up after Spica cast and subsequent additional Tübinger splint we performed anterior-posterior X-rays of the total pelvis after 1.5, 3, 5 and 10 years. Means are shown in Table 5. Moreover, 42/48 patients had at least one follow-up X-ray. Table 5 shows radiographic parameters of AI and CE angle during follow-up separated by the result of the last sonography. Figure 5 shows development of AI and CE-angle over time of follow-up. Factors influencing the outcome of congenital hip dysplasia were found at 3-year follow-up, but not at the others, perhaps due to loss to follow-up after 5 and 10 years. Abduction had a positive effect of the development of AI measured at 3 years age in our cohort (*p* = 0.044, beta = −0.609). No other positive effects were found to be statistically significant.

### 3.4. Complications

There were no complications such as avascular necrosis of the hip. 11.4% (11/96) hips had delayed remodelling of the femoral head. All showed a spherical femoral head at the end of individual follow-up.

## 4. Discussion

The aim of this study was to explore possible prognostic factors in spica cast MRI to predict the expected outcome. We included both hips, including ones without signs of DDH, in order to compare all evaluated hips irrespective of DDH severity in order to detect possible prognostic factors. Furthermore, spica cast and Tübinger splint treatment always involves both hips.

Published cohorts have ranged in size from 21–110 patients, allowing for a comparison with the results in this cohort of 49 patients [15,36,37,38,39,40]. The mean age at time of starting treatment in this study was 9.9 weeks and therefore far below most reported ages, which generally started treatment around 6 to 24 months of age or even older [20,36,37,38]. The poorer outcome and higher rates of AVN in some studies might be associated with older age at time of immobilization [39,41,42,43]. The absence of AVN in our population may also be due to short cast immobilization period compared to the literature and younger age of our patients [21,36,44,45].

During the follow-up, there was a reduction in the number of patients after the first radiological control up to the 10-year control. This was due to several reasons. Firstly, not all patients were able to complete the possible follow-up period. On the other hand, some parents refused further radiographs of their children after an unremarkable radiographic control or forget/neglected to attend the further appointments. Some may also have moved and/or are performing the checks somewhere else.

We achieved a good outcome compared to the literature, which may be a result of the younger patient age when initializing treatment. Furthermore, 1 of 49 patients underwent surgery and 3 of 49 had residual dysplasia requiring treatment during our follow-up (8.1% in total, 4/49), which is comparable to the literature, reporting unsuccessful closed reduction in 9–31.16% of cases [11,37,40,46]. The girl who underwent surgery had a normal history with treatment beginning at 6 weeks of age for an unstable 2c hip on the left side. After treatment with a spica cast and a Tübingen splint, a Graf type 1 hip was found on both sides. However, X-ray examinations showed an increasing recurrence on both sides during the follow-up, so that surgical hip reconstruction with derotation varisation shortening osteotomy and Dega acetabuloplasty was carried out. Further checks showed no further recurrence of DDH. The three others with residual dysplasia showed the following characteristics: One was pretreated externally with an abduction treatment and presented to our clinic at 22 weeks of age. After initial therapy, the subsequently prescribed abduction wedge was not tolerated at night. After 3 years, there was a persistent DDH on the left side, but no further presentation to our outpatient clinic. The second patient showed a right-sided recurrence of the DDH in the first X-ray control after successful initial treatment from the 8th week of life. After an abduction wedge was prescribed, she was lost to follow-up. In case of the last girl, after successful initial conservative therapy, a mild dysplasia was observed in the 5-year follow-up, which is still being observed.

To our knowledge, we are the first investigate possible prognostic factors on MRI after spica cast placement to predict the expected outcome. We measured several parameters which had been shown by Jaremko et al. to be reliable and evaluated several additional parameters which were considered to be of interest. CorAcet and Abduction had moderate reliability. AxAcet and AxPAcet were rated as unreliable, but were still considered of interest for this study. Pelvwidth was the only very reliable parameter Jaremko et al. had noted. This is more a parameter of age than of DDH, and in this study acted as an additional indirect test of a correlation between age at therapy initiation and outcome. Jaremko described other parameters studied as time-consuming or not clinically relevant [26], so we decided not to measure them. After casting, all hips were centered. Jaremko et al. described a large number of subluxated hips on MRI in their study, and half of the patients received open reduction or pelvic osteotomies before spica casting. In addition, the average age in their collective was 10 months (0–2 years), well above ours. These are only the main differences between the two groups of patients and illustrate the limitations of comparability between Jaremko et al. and our results.

In our patients, none of the MRI parameters showed a prognostic correlation with the sonographic outcome at the end of therapy. Despite MRI measurements, we also compared sonographic measurements and staging as well as age, sex, and radiographic parameters during follow-up. Here, we found that abduction had a positive effect of the development of AI at 3 years follow-up in our cohort (*p* = 0.044, beta = −0.609). This underlines the relevance of deep centering of the femoral head in the acetabulum for improved development of the hip. Care must be taken to retain the femoral head circulation by avoiding severe abduction.

An additional outcome influencing factor was the duration of cast therapy. The longer the cast was applied, the larger the alpha angle (*p* = 0.003, beta = 0.869).

None of the other measured values were shown to have an influence or predictive value on the expected outcome. This may be due to the relatively small cohort size and the small number of treatment failures. A larger cohort size may reveal that some of these other parameters do have prognostic value.

Limitations: this is a retrospective study without a control group. The relatively short follow-up of 47.4 months on average may have been too short to recognize further prognostic parameters.

## 5. Conclusions

In our cohort, we found the abduction angle to be the only influencing factor for improved development of the DDH. No other prognostic factors in MRI measurements, such as gender, age at time of the first spica cast, or treatment involving overhead extension were found to be predictive of mid-term outcome. Mild residual dysplasia in the first follow-up X-ray warrants further observation.

## Figures and Tables

**Figure 1 children-09-01010-f001:**
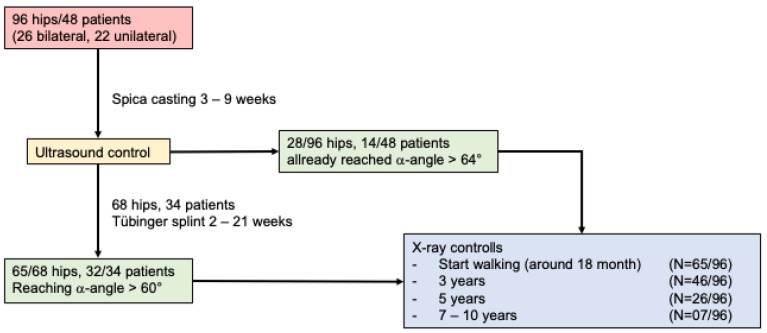
Shows the therapy algorithm of our treatment.

**Figure 2 children-09-01010-f002:**
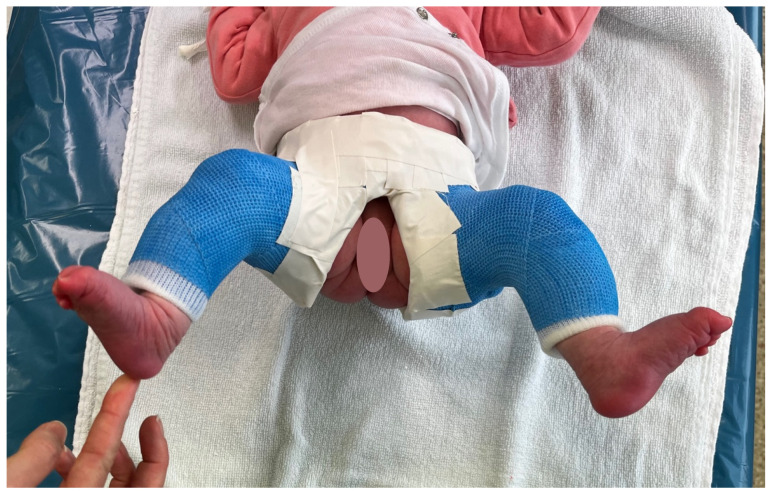
Shows a patient in a spica cast with hip abduction and flexion.

**Figure 3 children-09-01010-f003:**
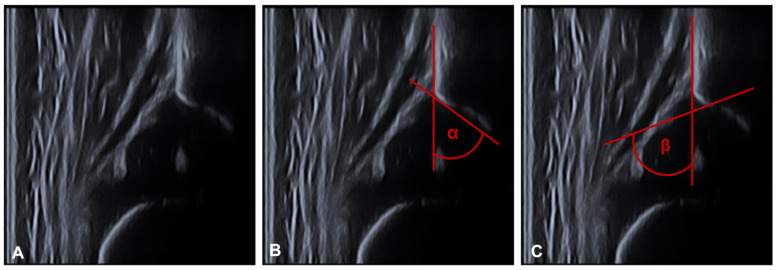
Shows the measurement of the α- and β-angle. (**A**) plain ultrasound image for anatomical identification and usability testing. (**B**) The tangent to the os ilium is the baseline for both angle determinations. For the α-angle, a tangent is drawn to the bony acetabulum starting from the lower edge of the os ilium (**C**) For the latter, a connecting line (display line) is drawn between the bony acetabular notch (turnover point) and the center of the acetabular labrum. The bony notch is located at the point where the acetabulum changes its profile from concave to convex. The angle between the two lines is called β-angle.

**Figure 4 children-09-01010-f004:**
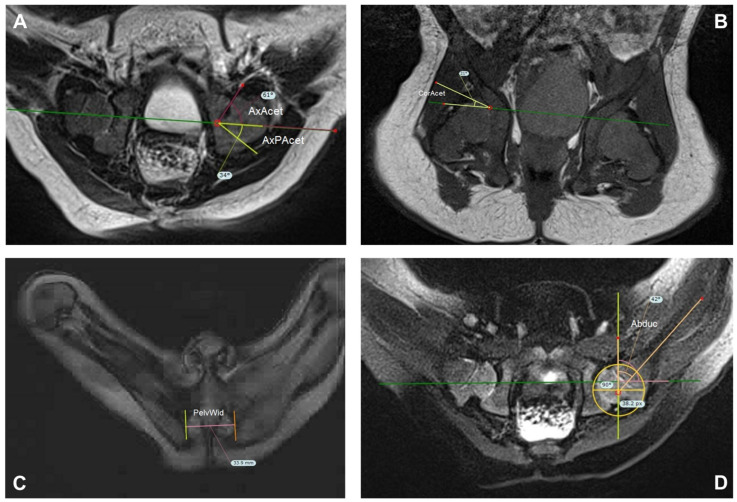
Overview of the measurement technic in the MRI described by Jaremko and performed in TraumaCad. (**A**) AxAcet/AxPAcet: Select the axial image for which the acetabular cup appears deepest. The bony anterior acetabulum index (red) is the angle is made between Hilgenreiner’s line (green) and a line joining the anterior edge of the bony acetabulum to the lateral edge of the triradiate cartilage. The bony posterior acetabular index (yellow) is measured similarly at the posterior joint line. (**B**) CorAcet: Select Slice were the acetabular roof is steepest. Angle (yellow) between Hilgenreiner‘s line (green) and a line joining the superior edge of the bony acetabulum to the lateral edge of the triradiate cartilage. (**C**) PelvWid: widest distance between medial ischial walls at the pelvic inlet, below the hip joints (**D**) Abduc: On the slice showing the largest diameter of the most normally positioned femoral bead, draw Hilgenrainer’s line (green) between anterior lateral edges of triradiate cartilage in the same fashion as on the coronal images. Abduc angle (orange) is the angle between the line along the mid femoral shaft and a perpendicular to Hilgenreiner’s line. (AxAcet AxPAcet = anterior/posterior bony axial acetabular angles, CorAcet = coronal acetabular angle, PelvWid = Pelvic width, Abduc = hip abduction angle) [26].

**Figure 5 children-09-01010-f005:**
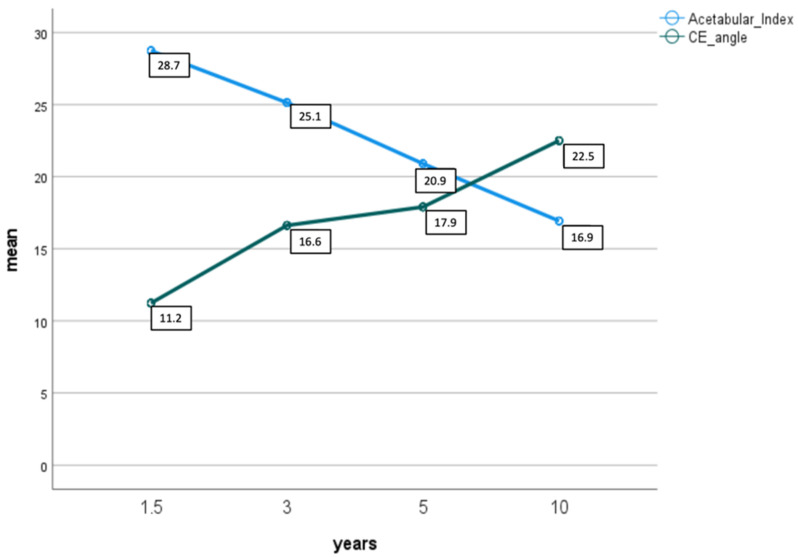
Shows mean development of acetabular index (light blue) and CE angle (green) during follow-up after 1.5, 3, 5 and 10 years.

**Table 1 children-09-01010-t001:** Post spica cast pelvic MRI protocol parameters, FOV field of view, fs fat saturation, PD proton density, TE echo time, TR repetition time.

Sequence	FOV (mm)	Voxel Size (mm)	Slice Thickness (mm)	Gap (%)	Foldover Direction	TR (ms)	TE (ms)	Acquisition Matrix
Cor PD tse 320	160	0.5 × 0.5 × 2.0	2.0	10	RL	1500	24	160
Cor T2 tseRB 320	200	0.6 × 0.6 × 2.0	2.0	10	FH	3180	97	200
Axial T2 tseRB fs 320	200	0.6 × 0.6 × 2.0	2.0	20	RL	2900	73	200
Axial T2 tseRB 320	200	0.6 × 0.6 × 2.0	2.0	20	RL	3400	97	200
Axial PD tseR 320	200	0.6 × 0.6 × 2.0	2.0	20	RL	1800	22	200
Sag PD tse 384 re/li	160	0.5 × 0.4 × 2.0	2.0	10	AP	1500	35	160

**Table 2 children-09-01010-t002:** In the ultrasound evaluation after spica cast therapy, 28 hips already showed an alpha angle of at least 64° and therefore did not require any further therapy. The following table shows the distribution of the initial Graf types of these patients.

Initial Graf Type	28 Hips Reached Alpha Angle of 64° or More after Spica Casting
1	10/28
2a	3/28
2c	4/28
3a	5/28
D	4/28
Unknown (no initial ultrasound)	2/28

**Table 3 children-09-01010-t003:** Shows ultrasound parameters at the beginning and end of therapy in total and separated by sex and age in weeks at time of beginning therapy. Alpha = alpha angle defined by Graf. Beta = beta angle defined by Graf.

		Mean Start Cast	Standard Deviation	Mean End Therapy	Standard Deviation
alpha	In total	52.2	9.8	65.6	3.3
	female	52.2	9.6	65.7	3.4
	male	51.8	11.4	64.5	1.8
beta	In total	74.2	9.3	63.9	6.8
	female	74.0	9.9	64.0	7.0
	male	75.1	7.9	63.7	5.8

**Table 4 children-09-01010-t004:** Summary of the magnetic resonance imaging (MRI) measurements in the first and last MRI. Abd = hip abduction angle. AxAcet = axial anterior acetabular angle. AxPAcet = axial posterior acetabular angle. CorAcet = coronal acetabular angle. PelvWid = pelvic width.

	Mean First MRI	Standard Deviation	Mean Second MRI	Standard Deviation
Abd	53.5 (39.3–66.2)	5.2	53.4 (29.0–62.1)	6.7
AxAcet	49.6 (32.8–68.2)	7.1	50.0 (32.8–62.4)	6.6
AxPAcet	46.8 (36.1–63.5)	6.0	46.8 (4.6–69.1)	9.3
CorAcet	25.6 (11.8–44.9)	6.6	24.2 (11.7–47.6)	7.8
PelvWid	35.8 (28.7–51.0)	5.0	38.2 (32.0–52.0)	5.1

**Table 5 children-09-01010-t005:** Shows radiographic measurements with mean and standard deviation of X-ray measurements of AI and CE angle after 1.5, 3, 5 and 10 years of follow-up separated by the result of the last sonography. CE-angle = centre edge angle by Wiberg. AI-angle = acetabular index. *n* = number.

Years		Excellent (α ≥ 64°)		Good (α < 64°)		Poor (α < 60°)	
		*n*		*n*		*n*	
1.5–2	AI	65	28.6 (s = 4.0)	10	29.5 (s = 4.2)	1	26.0
	CE		11.3 (s = 8.3)		8.8 (s = 8.8)		18.0
3	AI	46	25.1 (s = 4.8)	9	24.11 (s = 5.6)	1	22.0
	CE		16.4 (s = 5.6)		17.8 (s = 5.6)		21.0
5	AI	26	19.2 (s = 4.3)	8	21.5 (s = 5.2)	1	17.0
	CE		18.4 (s = 4.0)		18.38 (s = 3.8)		20.0
10	AI	7	16.3 (s = 3.9)	3	15.33 (s = 5.9)	0	
	CE		22.6 (s = 5.4)		24.33 (s = 4.7)		

## Data Availability

The data are not publicly available due to data privacy of the patients.

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
