# Peer review of "Outcome Prognostic Factors in MRI during Spica Cast Therapy Treating Developmental Hip Dysplasia with Midterm Follow-Up"

_children, 2022, doi:10.3390/children9071010_

Round 1

Reviewer 1 Report

1. Please expand on the indications of closed reduction . Is it correct to understand that closed reduction was performed in Graf 1 hips?

2. Is the spica treatment at such an early stage routine?

3. How was the duration of casting determined?.

4. Lines 98 and 99: what was the "therapy" ? (spica only or spica + subsequent splinting)

Author Response

Thank you very much for reviewing our manuscript. We hope that the explanations will make the individual points clearer. 

  1. Please expand on the indications of closed reduction . Is it correct to understand that closed reduction was performed in Graf 1 hips?

Graf hips Typ 1 were only "treated" if the patient was affected by congenital hip dysplasia on the opposite side, since a spica cast always involves both sides. Reduction was not necessary or carried out on the unaffected side. We have tried to clarify this part in 2.1.

  1. Is the spica treatment at such an early stage routine?

In Germany, there is an obligatory hip ultrasound screening according to Graf in the 6-8 weeks after birth. If risk factors for a DDH are present, this is even done on the 2-3 day after birth If DDH is detected, the appropriate treatment is started. Even though this is rather early in an international comparison, we have had good experience with an early start of treatment and thus shorter treatment time according to the Tschauner maturation curve. Both the children and parents benefit from a shortest possible casting period, since the spica cast makes hygiene more difficult.

  1. How was the duration of casting determined?

The duration of the cast therapy was determined according to the patient's age and the severity of the disease. In general, the standard therapy was two casts, each left on for 3 weeks. Afterwards, an ultrasound check was carried out first. If the DDH was still severe, a further spica cast was applied for an additional 3 weeks if necessary. If DDH was still mild, a Tübingen splint was applied, but if an alpha angle of 64° or more was already measured, the therapy was also finished. If severe DDH was initially present in very young infant, an ultrasound control may have been carried out after the initial 3-week cast therapy before deciding whether to place another spica cast on for the second three weeks of treatment.

We have added this supplementary statement accordingly in the manuscript.

  1. Lines 98 and 99: what was the "therapy" ? (spica only or spica + subsequent splinting)

After the planned spica cast therapy, an ultrasound check was carried out. If the alpha angle was still below 60°, the therapy was continued with a Tübingen splint until the desired post-maturation of the hip occurred.

We have added this supplementary statement accordingly in the manuscript.

Reviewer 2 Report

Need extensive restructuring of English grammar and style first- recommended to grammatically edit and resubmit 

Author Response

Thank you for reviewing our manuscript. It was revised by a native speaker.

Reviewer 3 Report

First of all, I would like to congratulate the authors on their research.

My comments:

In the abstract, lines 21-22, mention of Jaremko et al. kind of seems like a citation (even though it is not) and should not be included in this form in the abstract. Try to rephrase.

In the introduction, this part should be a little longer, try to cover broad aspects of diagnostics.

Line 31 how can DDH be detected at an early age? Try to include only a short section/paragraph on epidemiology, also on risk factors and diagnostics at an early age (as this is a clinical study not a molecular, I leave the inclusion of genetics /candidate genes to the authors' own decision, as it is not necessary).

Examples of recently published works within publishers' portfolios:

Epidemiology study https://doi.org/10.3390/ijerph18126589

Risk factors https://doi.org/10.3390/medicina56040153

For mention of indices by Jaremko et al in the introduction, include a section pointing to material and methods for further detail, as this is the second mention after the abstract (needs rephrasing) and no information is given.

Material and methods

Was the decision not to use anesthesia problematic in MR imaging? Or were these examinations without any problems due to little movement in the treated hip?

In statistical analysis reference on SPSS needs revision.

Results

Line 178 “In total mean of treatment duration” needs rephrasing. Proposition: The mean duration of treatment ...

Line 183-184 results in this format have bad reading clarity, try to remake it in a small table or another way to improve on readability. Also, counts of patients don’t add up to 28, please check the numbers.

Line 188+ These 3 patients with residual dysplasia, can you identify the parameters of these patients? If either a worse initial examination or surgery, ….

Line 227 Inconsistency with material and methods as first only one patient was mentioned as not having an initial USG, now there are two. Please check the facts within the whole article.

Discussion

Line 262 for clarity and presentation better state that only one of your patients needed surgery, not in percentual value. Or only one (8.1%) of 49 patients.

Line 286 needs rephrasing /reordering words for better understanding for readers

Lines 291-292 few examples of questions not being answered

Line 293+ This paragraph needs reworking for clarity

P.S. I suppose, parts with “Error! Reference source not found“ are up to the editorial team to address.

Author Response

  1. In the abstract, lines 21-22, mention of Jaremko et al. kind of seems like a citation (even though it is not) and should not be included in this form in the abstract. Try to rephrase.

Thank you for pointing this out. We have changed the passage as follows:

The following parameters were measured in the MRI: hip abduction angle, coronal, anterior and posterior bony axial acetabular angles and pelvic width.

  1. In the introduction, this part should be a little longer, try to cover broad aspects of diagnostics.

Thank you for pointing this out. We have added several additions regarding the diagnostics for DDH in the introduction and marked them accordingly in yellow.

  1. Line 31 how can DDH be detected at an early age? Try to include only a short section/paragraph on epidemiology, also on risk factors and diagnostics at an early age (as this is a clinical study not a molecular, I leave the inclusion of genetics /candidate genes to the authors' own decision, as it is not necessary).

Thank you for pointing this out. We have made several additions regarding the epidemiology and risk factors of DDH to the introduction and marked them accordingly in yellow.

  1. For mention of indices by Jaremko et al in the introduction, include a section pointing to material and methods for further detail, as this is the second mention after the abstract (needs rephrasing) and no information is given.

We have made the recommended change (see chapter 2.3 measurements) on line 73

  1. Was the decision not to use anesthesia problematic in MR imaging? Or were these examinations without any problems due to little movement in the treated hip?

Not having any more anesthesia in the MRI was not problematic. We took advantage of the residual anesthesia directly postoperatively and lightly fixed the cast with a sandbag in rather unsteady infants. This minimized movement artefacts. In addition, the mother was also in the MRI room with her infant to calm it.

  1. In statistical analysis reference on SPSS needs revision.

Thank you for pointing this out. We corrected it to the newest version 28 which was used for the statistics.

  1. Line 178 “In total mean of treatment duration” needs rephrasing. Proposition:The mean duration of treatment ...

The whole article has been revised by a native speaker.

  1. Line 183-184 results in this format have bad reading clarity, try to remake it in a small table or another way to improve on readability. Also, counts of patients don’t add up to 28, please check the numbers.

Table 2 was inserted for clarification.

  1. Line 188+ These 3 patients with residual dysplasia, can you identify the parameters of these patients? If either a worse initial examination or surgery, ….

The girl who underwent surgery had a normal history with therapy beginning at 6 weeks of age with an unstable 2c hip on the left side. After a spica cast and a Tübingen splint, a type 1 hip according to Graf was found on both sides. However, X-ray examinations showed an increasing recurrence on both sides during the follow-up, so that surgical hip reconstruction with derotation varisation shortening osteotomy and Dega osteotomy was recommended. Further checks revealed a normal hip roof. The 3 others with residual dysplasia showed the following characteristics. One was pretreated externally with a abduction treatment and presented to us at 22 weeks of age. After initial therapy, the abduction wedge was not tolerated at night. After 3 years, there was a persistent DDH on the left side, but no further presentation to our outpatient clinic. The second one showed a right-sided recurrence of the DDH in the first X-ray control after successful initial treatment from the 8th week of life. After an abduction wedge was prescribed, she was lost to follow up. In case of the last girl, after successful initial conservative therapy, a mild dysplasia was observed in the 5-year follow-up, which is still being observed.

  1. Line 227 Inconsistency with material and methods as first only one patient was mentioned as not having an initial USG, now there are two. Please check the facts within the whole article.

It is one patient, but two hips. This has been clarified in the article now.

  1. Line 262 for clarity and presentation better state that only one of your patients needed surgery, not in percentual value. Or only one (8.1%) of 49 patients.

We restructured the phrase for clarification: One of 49 patients underwent surgery and 3 of 49 had residual dysplasia requiring therapy during our follow up (8.1% in total, 4/49).

  1. Line 286 needs rephrasing /reordering words for better understanding for readers

The article has been revised by a native speaker. We hope that it is now easier to understand.

  1. Lines 291-292 few examples of questions not being answered

It is possibly due to the relatively short follow-up that no further prognostic parameters could be found.

  1. Line 293+ This paragraph needs reworking for clarity

The whole article was revised by a native speaker. Hopefully it is clearer now.

  1. S. I suppose, parts with “Error! Reference sourcenot found“ are up to the editorial team to address.

There was a formatting error for the figures and tables. We corrected this issue. Now it should be fine.

Round 2

Reviewer 1 Report

Congratulations for this work and thank you for the revisions, they are most welcome. Your subject is very interesting. I agree that a prospective study with large numbers is necessary clarify the role of the MRI scan in the treatment of DDH.

Author Response

Thank you for re-reviewing our work. We appreciate your positive comments and your interest in the topic.

Reviewer 2 Report

Questions

  • How were the follow up time intervals selected? 
  • In section 2.3, would recommend a figure since you are describing in such detail the alpha and beta angles - would help to clarify how to reproduce for the readers  (line 172)
  • How many patients had their femoral heads’ ossify and therefore it was necessary to switch to XR from US? (Line 177)
  • What is the significance of line 241-42? Would clarify 
  • For weaknesses- there is a significant drop off in follow up after 3 years and then even more at 5 years- would comment on this 

Grammar edits

  • Line 18- treatment began- not treatment was begun
  • Line 19- the children were followed for - not the children were follow-up for 
  • Line 26- should be outcomes
  • Line 33- should be: varies from, not varies at 
  • Line 54- should be AP not a.p.
  • Line 81- sentence starting with looking is not a complete sentence 
  • Line 99- the number 1 should be spelled out at the beginning of the sentence as well as for the number 6 
  • Line 132- should be “a” before Tubing splint
  • Line 211- should be a space after 18 and month should be months 
  • Line 225- data is given, not data are given 
  • General rule: if the number is less than 10- do not sure number itself, write out the word itself 
  •  

Author Response

Thank you for reviewing our manuscript. We added the following points you noted to our manuscript.

1. How were the follow up time intervals selected? 

Children with DDH should be followed up radiologically until completion of growth, due to possibility of late recurrence. If no signs of DDH are present during follow-up, we still recommend follow-up at walking age, after 8-10 years and after the end of growth. If DDH is apparent on follow-up, the time intervals between follow-ups are shortened, and treatment initiated if deemed necessary. In our cases, severe DDH was present, so spica cast therapy was necessary and therefore the control intervals were reduced to walking age, again after 3 and 5 years, and after 10 years. Due to the study period and follow-up time frame, no patient has reached skeletal maturity at this point.

2. In section 2.3, would recommend a figure since you are describing in such detail the alpha and beta angles - would help to clarify how to reproduce for the readers  (line 172)

We have added a figure (Figure 3) to clarify this.

3. How many patients had their femoral heads’ ossify and therefore it was necessary to switch to XR from US? (Line 177)

Femoral head ossification usually begins around 8 months of age. 6 children were not seen at the first radiological follow-up time point after the start of walking age. All other children had femoral head ossification by the follow-up time.

4. What is the significance of line 241-42?

The significance was p=0.014. We hope this makes it clearer.

5. For weaknesses- there is a significant drop off in follow up after 3 years and then even more at 5 years- would comment on this 

During the follow-up, there was a reduction in the number of patients after the first radiological control up to the 10-year control. This was due to several reasons. Firstly, not all patients were able to complete the possible follow-up period. On the other hand, some parents refused further radiographs of their children after an unremarkable radiographic control or forget/neglected to attend the further appointments. Some may also have moved and/or are performing the checks somewhere else.

6. Language corrections line 18-225

Thank you for pointing this out. We corrected the language according to your suggestions.

Reviewer 3 Report

I congratulate the authors on a quick rewriting of the article.

Except for the introduction all my questions and remarks have been sufficiently addressed and corrected, so I have no more comments on these sections. Also, English narrating and style are much more cohesive and readable now.

This is a minor issue, but an important one nonetheless. This is also aimed at future research by authors.

I only have one more comment regarding old and newly added citations for epidemiology and risk factors, etc. (1-7 and further), of which many are 15-20 years old, which for reviews is a pretty long time. Due to this, in my first report, I proposed 2 model articles (I found them by typing DDH in the mdpi.com webpage search bar) from the publishers' portfolio, which are recent and should serve the authors as examples. Inclusion of these two specific articles was not mandatory for my approval, but citing more recent research/review articles - best last 5 (max 10) years is. Only use older articles if these contain fundamental information or a cornerstone in a given subject or field, and not to hurt anyone, 20 years old reviews rarely do. This practice helps promote newer articles, newer facts/information, and new takes on an issue, and also citing articles from the publishers' journals is a professional courtesy.

Main takeaway: Cite newer research and newer reviews if possible. Don't cite old articles for the sake of citations, if these aren't fundamental in the field in which your work is based.

Author Response

  1. Except for the introduction all my questions and remarks have been sufficiently addressed and corrected, so I have no more comments on these sections. Also, English narrating and style are much more cohesive and readable now.

We have updated the references.

Unfortunately, we could not open or display the linked sources, so we had to choose other ones.